# NATURE: Natural Auxiliary Text Utterances for Realistic Spoken Language Evaluation

**David Alfonso-Hermelo**
Huawei Noah's Ark Lab
david.ah@huawei.com

**Ahmad Rashid**
Huawei Noah's Ark Lab
ahmad.rashid@huawei.com

**Abbas Ghaddar**
Huawei Noah's Ark Lab
abbas.ghaddar@huawei.com

**Philippe Langlais**
RALI/DIRO, Université de Montréal
felipe@iro.umontreal.ca

**Mehdi Rezagholizadeh**
Huawei Noah's Ark Lab
mehdi.rezagholizadeh@huawei.com

## Abstract

Slot-filling and intent detection are the backbone of conversational agents such as voice assistants and they are active areas of research. Even though state-of-the-art techniques on publicly available benchmarks show impressive performance, their ability to generalize to realistic scenarios has yet to be improved. In this work, we present NATURE, a set of simple spoken language oriented transformations, applied to the evaluation set of datasets, to introduce human spoken language variations while preserving the semantics of an utterance. We apply NATURE to common slot-filling and intent detection benchmarks and demonstrate that simple deviations from the standard test set by NATURE can deteriorate model performances significantly. Additionally, we apply different strategies to mitigate the effects of NATURE and report that data-augmentation leads to some improvement.

## 1   Introduction

The growing demand for Virtual Assistant systems (Uğurlu et al. (2020), Li et al. (2021)) has led to advances in conversational and spoken language oriented models, Natural Language Understanding (NLU), and Spoken Language Understanding (SLU). One of the backbones of NLU and SLU is the joint tasks of Intent Detection (ID, identification of the speaker's intent) and Slot-filling (SF, extraction of the semantic constituents from the utterance). In recent years, NLU models specialized in ID and SF have obtained outstanding results (Qin et al. (2019), Wang et al. (2018), Yamada et al. (2020)). However, these models usually lack satisfying generalization capabilities (McCoy et al. (2019), Gururangan et al. (2018), Balasubramanian et al. (2020), Lin et al. (2020)).

Data Augmentation (DA) is one of the well-known solutions to this problem (Hou et al. (2020), Louvan and Magnini (2020), Kale and Siddhant (2021)). However, without looking at the test set, we cannot account for all the patterns which are missing in the training set. Moreover, it still does not resolve the issue of a lack of generalization to out-of-distribution evaluation. This is an issue in real scenarios, specially considering the paraphrase richness of spoken language. Other works propose modified evaluation sets (Lin et al. (2020), Agarwal et al. (2020)). This is a valid option but for some tasks (as ID and SF) the available data is scarce, rarely open-source and producing new and qualitative data is labor-intensive, time-consuming, and expensive.

We propose a framework that focuses on transforming the existing test sets by applying simple, spoken language-oriented, realistic operators that slightly modify the input sentence but without

Submitted to the 35th Conference on Neural Information Processing Systems (NeurIPS 2021) Track on Datasets and Benchmarks. Do not distribute.

| Utterance | Task: | Model Prediction Errors |
|---|---|---|
| play party anthems
→ **ploy** party anthems | ID: | `Play_Music`
→ Search_Creative_Work |
| play some sixties music
→ **plays** some sixties music | SF: | [sixties]:`year`
→ [sixties]:`year`; [plays]:`album` |
| listen to dragon ball: music collection
→ **like** listen to dragon ball: music collection | ID: | `Search_Creative_Work`
→ Play_Music |
| | SF: | [dragon ball: music collection]:`object_name`
→ [dragon ball]:`artist`; [collection]:`album` |

Figure 1: Examples of NATURE-altered utterances with badly predicted slots and or intent. The altered utterance is preceded by a →.

altering the original meaning (as we shall see). By realistic, we mean that modified utterances remain semantically similar to the original ones. We call this framework NATURE (*Naive Alterations of Textual Utterances for Realistic Evaluation*). Figure 1 shows examples of altered utterances where a state-of-the-art model (Qin et al. (2019)) correctly predicted the label for the original utterance but failed for the altered utterance.

We conduct experiments that apply our framework to standard benchmarks and compare the *before and after* performances of state-of-the-art models. The results illustrate the heuristic dependencies of each model.

## 2 Related Work

### 2.1 Realizing model use shortcuts

A growing number of studies identify a tendency in NLU models to leverage the superficial features and language artifacts instead of generalizing over the semantic content. A naive way to force generalization is to automatically add noise to the training set, however, as demonstrated by Belinkov and Bisk (2017), models trained on synthetic noise do not necessarily perform well on natural noise, requiring a more elaborated approach. Given our incapacity to control what features these models learn, each task requires an in-depth analysis and a data or model modification that guides it to the correct answer. For the political claims detection task Padó et al. (2019) and Dayanik and Padó (2020) unveil a strong bias towards the claims made by frequent actors that require masking the actor and its pronouns during training to improve the performance. Other works (Gururangan et al. (2018), Poliak et al. (2018), Zellers et al. (2018), McCoy et al. (2019), Naik et al. (2018)) have focused on the artifact and heuristic over-fitting for the Natural Language Inference (NLI) task or for the Question-Answering (QA) task (Jia and Liang (2017)). The work of Balasubramanian et al. (2020) show how substituting Named-Entities (NEs) influence the robustness of BERT-based models for different tasks (NLI, co-reference resolution and grammar error correction). To the best of our knowledge, no work has attempted to demonstrate that the benchmarks and models for the dual tasks of SF and ID rely on frequent heuristic patterns.

### 2.2 Alternative evaluation

Some researchers have proposed evaluation sets with naturally occurring adverse sentences for different tasks such as HANS for MNLI (McCoy et al. (2019)) or PAWS( Zhang et al. (2019)) and PAWS-X (Yang et al. (2019)) for paraphrase identification. Another strategy involves a systematic alteration of the test set (Lin et al. (2020)). This has gained popularity in recent years with a growing interest in more challenging and adversarial evaluation frameworks. However, a more challenging test set has to ensure high quality annotation, which is why many papers have suggested an human-in-the-loop approach (Kaushik et al. (2019), Gardner et al. (2020), Kiela et al. (2021)). But these approaches are costly, specially due to the number and quality of annotators necessary to

produce a high-quality output. Generalization is more easily achieved when the training data is large and diverse. A model can be effective, yet, if it is only fed with small and/or similar data, it will have difficulties to achieve robustness. Some researchers (Louvan and Magnini (2020), Zeng et al. (2020), Dai and Adel (2020), Min et al. (2020), Moosavi et al. (2020)) use DA strategies to improve the training data and help boost a model's performance.
Other researchers have taken a different path and suggest a whole different way of evaluating: testing multiple task-agnostic requisites instead of using a test set that matches the train and validation sets (Ribeiro et al. (2020), Goel et al. (2021)).

## 2.3 Test set alteration methods

There has been many proposals of spoken-language oriented alteration methods (Tsvetkov et al. (2014), Simonnet et al. (2018), Li et al. (2018), Gopalakrishnan et al. (2020)) but the ones we are interested in require to change the utterance form while maintaining the original semantic value of each token (in the form of labels). Very few works have managed to devise methods that change the form while maintaining the semantic labeling, such as the work of Yin et al. (2020) where the authors suggest altering methods that emulate non-native errors or the work of Li et al. (2020) where they use simple methods to produce more counterfactual versions of the original utterances.

# 3 Methodology

In this section we describe the operators used to generate new utterances out of a given one. We present examples for each operator on Figure 2.

## 3.1 Fillers

Fillers are ubiquitous in everyday spoken language and often appear in human-to-human dialog (transcribed to text) corpora (such as the Switchboard corpus Godfrey et al. (1992), composed of approximately 1.6% fillers Shriberg (2001)). Yet they are intentionally cleaned off in SF and ID benchmarks. Fillers serve as hesitation markers (e.g.: *Bring me the, **like**, Greek yogurt. I've heard it's really, **you know**, savoury.*) or as introduction/closure of a turn of speech (e.g., ***Now,** bring me the Greek yogurt **please and thank you**. **Actually,** I've heard it's really savoury, **right?***). Fillers are semantically poor and do not add essential information, and therefore, do not change the overall meaning of an utterance.

We propose 4 different filler operators:

- **Begin-of-sentence** (BOS): a small introductory filler phrase at the beginning of the utterance, such as: *so*, *like*, *actually*, *okay so*, *so okay*, *so basically*, *now* or *well*.

- **End-of-sentence** (EOS): a small conclusive filler phrase at the end of the utterance, such as: *if you please*, *please*, *pretty please*, *please and thank you*, *now please*, *if you can*, *now*, *right now*, *right away*, *right this minute*, *will you ?*, *would you ?*, *can you ?*, *would you mind ?*.

- **Pre-verb**: a filler word or sequence of words appearing before the utterance's verb or verbal phrase, such as: *like*, *basically* or *actually*.

- **Post-verb**: a filler word or sequence of words appearing after the utterance's verb or verbal phrase, such as: *basically*, *actually*, *like* or *you know*.

BOS and EOS operators simply add a filler at the very beginning or the end of the utterance, respectively. The pre-verb and post-verb operators require us to find the part-of-speech (POS) tag of the utterance tokens (we use the NLTK library to find the POS of the tokens). Then the filler is placed at the correct place. We add a fail-safe rule to ensure that a filler is added if no verb is found where expected. To that end, we use the overly-recurrent filler, *like*, and the first appearing Named Entity as a pivot instead of the first appearing verb e.g., *let's check **like** avengers*).

| Test set | Example sentence |
|---|---|
| Original | add tune to sxsw fresh playlist |
| BOS Filler | **okay so** add tune to sxsw fresh playlist |
| Pre-V. Filler | **like** add tune to sxsw fresh playlist |
| Post-V. Filler | add tune **actually** to sxsw fresh playlist |
| EOS Filler | add tune to sxsw fresh playlist **if you can** |
| Synonym V. | **play** tune to sxsw fresh playlist |
| Synonym Adj. | add tune to sxsw **cool** playlist |
| Synonym Adv. | add **prior** tune to sxsw fresh playlist |
| Synonym Any | **mix** tune to sxsw fresh playlist |
| Synonym StopW | add tune **the** sxsw fresh playlist |
| Speako | add **tua** to sxsw fresh playlist |

Figure 2: Processed variants of original utterances from the SNIPS corpus. The tokens labeled as *music_item* appear with a dotted underline and the tokens labeled as *playlist* show a dashed underline. In SNIPS, the *sxsw* token is part of a playlist name and an abbreviation of *South by Southwest*.

| Token in context | Wiktionary synonyms | BERT candidates |
|---|---|---|
| let me buy it 
 verb | purchase, accept, [...] | get, buy, present, make, purchase, offer, give, sell, [...] |
| is it large ? 
 adj | giant, big, huge, [...] | unusual, big, dangerous, large, powerful, [...] |
| i said it quickly 
 adv | rapidly, fast | fast, well, strong, high, good, deep, large, slow, [...] |
| give me freedom 
 noun | liberty, license, [...] | rights, property, freedom, status, goods, liberty, [...] |
| i found the ball 
 stopword | le | the, second, also, third, their, still, a, our, 2nd, [...] |

Figure 3: Target words (underlined) of various POS and their synonyms taken from the crowd-sourced dictionary Wiktionary and candidates obtained using a pre-trained BERT language model.

## 3.2 Synonymy

A synonym is a word that can be interchanged with another in context, without changing the meaning of the whole. To replicate this semantic operation, we select the POS corresponding to our operator (among verb, adjective, adverb, etc.). We then select a word of that type in the input utterance and make a list of potential synonym candidates (with the same POS tag) to replace it. Then we select the most probable of the candidates as our replacement. We use the pre-trained BERT-base model with a Language Modeling head on top to produce the synonym candidates instead of a human populated dictionary (such as Wiktionary) since not all dictionary entries show synonyms. We first randomly choose a POS tag and find a target token which has this tag in our utterance. Then we replace the target with a special [MASK] token. We feed this utterance into BERT and obtain a list of candidates from most to least probable.

In case the sentence contains no token with the target POS, we use the more common *noun* POS. We observe an example in the *Syn. Adv.* row in Table 2.

As we can see in Figure 3, not all BERT candidates are suitable synonyms of the target token. We remove candidates that do not have the same POS of the target token. For a better performance, we put each candidate in the context of the utterance before extracting candidate POS. We have 5 different Synonymy operators based on different target POS: **verb**, **adjective**, **adverb**, **any** (at random between verb, adjective, adverb or noun), **stop-words** (grammatical and most common words).

## 3.3 Speako

Some words sound similar to others but have a different meaning altogether (e.g., *decent* and *descent*). This operator is based on the idea that anyone can make an error, but an efficient and robust model

should be able to recover a minor mistake using the context. Thus, we introduce speakos (slip of the tongue, speech-to-text misinterpretation), which are common in user-machine communication.

To do so, we use a prepared dictionary of tokens appearing 1000+ times in the whole English Wikipedia[1]. We convert each entry of the dictionary into its representation in International Phonetic Alphabet (IPA). We randomly select one token from the sentence, convert it to IPA, calculate the similarity between it and the dictionary's entries (using Levenshtein distance) and replace it with the closest candidate. For instance, the sentence *let me watch* *(/wɑtʃ/) a comedy video* could be transformed into *let me which* *(/wɪtʃ/) a comedy video*).

# 4 Experimental Setup

## 4.1 Data

In our work, we use 3 popular open-source benchmarks [2] which are summarized in Table 1:

**Airline Travel Information System (ATIS)** [3] Hemphill et al. (1990) introduced an NLU benchmark for the SF and ID tasks with 18 different intent labels, 127 slot labels and a vocabulary of 939 tokens. It contains annotated utterances corresponding to flight reservations, spoken dialogues and requests.

**SNIPS** [4] Coucke et al. (2018) proposed the SNIPS voice platform, from which a dataset of queries for the SF and ID tasks with 7 intent labels, 72 slot labels and a vocabulary of 12k tokens were extracted.

**NLU-ED** [5] is a dataset of 25K human annotated utterances using the Amazon Mechanical Turk service Liu et al. (2019). This NLU benchmark for the SF and ID tasks is comprised of 69 intent labels, 108 slot labels and a vocabulary of 7.9k tokens.

Following the common practice in the field (Hakkani-Tür et al. (2016), Goo et al. (2018), Qin et al. (2019), Razumovskaia et al. (2021), Krishnan et al. (2021)), we report the performance of SF using the F1 score. Moreover, we propose an End-to-End accuracy (E2E) metric (sometimes referred in the literature as the sentence-level semantic accuracy (Qin et al. (2019))). This metric counts true positives when all the predicted labels (intent+slots) match the ground truth labels. This allows us to combine the SF and ID performance in a single more strict metric.

| Benchmark | | Train | Valid. | Test |
|---|---|---:|---:|---:|
| ATIS | Sent | 4 478 | 500 | 893 |
| | Words | 50 497 | 5 703 | 9 164 |
| | Voc | 867 | 463 | 448 |
| SNIPS | Sent | 13 084 | 700 | 700 |
| | Words | 117 700 | 6 384 | 6 354 |
| | Voc | 11 418 | 1 571 | 1 624 |
| NLU-ED | Sent | 20 628 | 2 544 | 2 544 |
| | Words | 145 950 | 18 167 | 17 347 |
| | Voc | 7 010 | 2 182 | 2 072 |

Table 1: Dataset size information of our benchmarks: ATIS, SNIPS and NLU-ED.

Any dialog-based dataset extracted from real user situations has the potential of containing private and security sensitive information. This is the main cause for the relative low amount of datasets for

---

[1] We empirically observed that removing all tokens that had a co-occurrence lower than 1000 eliminated most of the nonsensical strings and extreme misspellings and conserved most functional words and very common typos.

[2] We did not select the SGD dataset of Rastogi et al. (2020) despite being recent and large, since it is a multi-turn dialog benchmark and cannot be used out of the box for the SF and ID tasks.

[3] CGNU General Public License, version 2

[4] Creative Commons Zero v1.0 Universal License

[5] Creative Commons Attribution 4.0 International License

SF and ID. The benchmarks we mention are well known and cautiously cleaned (as presented in Section 3). Our operators purposely avoid using any type of resource that would contain personal information. To the best of our knowledge, our work is not detrimental to people's safety, privacy, security, rights or to the environment in any way.

## 4.2 Models

We use two different state-of-the-art models:

**Stack-Prop+BERT** (Qin et al., 2019) uses BERT as a token-level encoder that feeds into two different BiLSTMs, one per each task. The output of the SF BiLSTM is added to the ID BiLSTM input in order to produce a token-level intent prediction which is further averaged into a sentence-level prediction.

**Bi-RNN** (Wang et al., 2018) uses two correlated BiLSTMs that cross-impact each other by accessing the other's hidden states and come to a joint prediction for ID and SF.

The pre-trained version of these models were not available[6]. For ATIS and SNIPS, we trained the models using the same hyperparameters proposed in the documentation by Qin et al. (2019)[7] and Wang et al. (2018)[8], respectively. For NLU-ED, we use the hyperparameters from SNIPS, as their size is comparable. Our trained models obtained comparable results to their published counterpart (see in Appendix). To train the models, we used 1 NVIDIA Tesla V100 with 32Gb of internal memory. It took between 3 and 71 hours to train the Stack-Prop+BERT model (Qin et al., 2019) (depending on the size of the benchmark), and between 68 and 130 hours to train the Bi-RNN model (Wang et al., 2018).

## 4.3 Modified NATURE Test Sets

Since the original test sets only cover a limited set of patterns, we transform them by applying our NATURE patterns to obtain test sets of the same size as the original ones. As previously illustrated, NATURE operators offer simple ways of altering utterances. In order to avoid rendering utterances unrecognizable from their original version, we only apply one operator at a time and only once in the sentence (e.g. we add 1 filler or synonymize one token or transform a token into its speako version). We design 2 NATURE experimental test sets: *Random* and *Hard*. In the Random setting, for each utterance, we apply one operator at random. **This random selection may cause an unbalanced distribution of alterations (some operators being more used that others). To obtain a more impartial score, we repeat the random operator selection 10 times and calculate the mean score**.
For the Hard setting experiments, after applying all our operators on each utterance and gathering all candidates, we use a relatively simple BERT-based model to calculate the performance of each candidate. We use JointBERT [9], which is an unofficial implementation of the SF and ID architecture described in Chen et al. (2019) to extract (for each utterance) the candidate that performs more harshly. The assumption being that the candidate that performed poorly for one model will have a greater chance of performing poorly on other models.
The Random test set is meant to show how a random small change in the sentence can influence evaluation while the Hard test set is meant to assess the lower-bound performance of how much the model depends on similar pattern sentences to obtain the correct prediction.

---

[6] https://github.com/LeePleased/StackPropagation-SLU and https://github.com/ray075hl/Bi-Model-Intent-And-Slot

[7] 300 epochs, 0.001 learning rate, 0.4 dropout rate, 256 encoder hidden dimensions, 1024 attention hidden dimensions, 128 attention output dimensions, 256 word embedding dimensions for ATIS and 32 for SNIPS.

[8] 500 epochs, max sentence length of 120, 0.001 learning rate, 0.2 dropout rate, 300 word embedding size, 200 LSTM hidden size

[9] https://github.com/monologg/JointBERT

| Operator | ATIS | SNIPS | NLU-ED |
|---|---|---|---|
| BOS Filler | 0.8 | 0.1 | 2.5 |
| Pre-V. Filler | 6.0 | 3.7 | 16.0 |
| Post-V. Filler | 1.9 | 8.6 | 5.1 |
| EOS Filler | 9.0 | 52.3 | 8.3 |
| Syn. V. | 25.6 | 5.4 | 16.3 |
| Syn. Adj. | 29.2 | 15.0 | 23.4 |
| Syn. Adv. | 11.8 | 5.6 | 10.2 |
| Syn. Any | 5.3 | 1.1 | 4.8 |
| Syn. StopW | 3.2 | 2.7 | 6.4 |
| Speako | 7.2 | 5.4 | 6.9 |

Table 2: Distribution of JointBERT-selected operators for the Hard experimental test set.

## 4.4 Augmented Training Sets

Even though our NATURE operators are designed for different purposes, some of these operators may look like certain DA strategies. However, in this subsection, we show to what extent our current operators are different from most famous heuristic DA techniques. In this regard, we apply standard DA strategies to the train and validation sets and illustrate their impact on the model's generalization ability. We use common automatic DA strategies from the NLPaug library (Ma, 2019) that allow to easily relabel the augmented data using the original labels:

1. **Keyboard Augmentation**: simulates keyboard distance error.
   (e.g, *find a tv seriSs called armaRdvdon summer*)

2. **Spelling Augmentation**: substitutes word according to spelling mistake dictionary.
   (e.g., *fine a tv serie called armageddon summer*)

3. **Synonym Augmentation**: substitutes similar word according to WordNet/PPDB synonym.
   (e.g., *find a tv set series called armageddon summertime*)

4. **Antonym Augmentation**: substitutes opposite meaning word according to WordNet antonym.
   (e.g., *lose a tv series called armageddon summer*)

5. **TF-IDF Augmentation**: uses the TF-IDF measure to find out how a word should be augmented.
   (e.g., *find tv series called armageddon forms*)

6. **Contextual Word Embeddings Augmentation**: feeds surroundings word to BERT, DistilBERT, RoBERTa or XLNet language model to find out the most suitable word for augmentation.
   (e.g., *find a second series called armageddon ii*)

We apply the DA strategies exclusively to the train and validation sets, choosing 1 of the 6 DA functions at random and adding one output to the original dataset which will give us a training and validation data twice as large as the original training and validation sets. One might notice that some of the DA techniques implemented in this toolkit are close in nature to some of our NATURE operators, still (as we shall see) this DA toolkit does not suffice to generalize well to the transformations of NATURE.

## 5 Results and Discussion

### 5.1 Qualitative Evaluation

Our assumption is that the operator-generated utterances share the same meaning and labeling as the original sentence. In order to measure this, we conducted a small but representative multiple-choice survey. We select 120 operator-altered utterances from the ATIS, SNIPS and NLU-ED benchmarks. We selected at random 40 utterances from each benchmark, making sure they were also evenly

distributed between operators (12 utterances per operator). In addition to these, we cherry-picked 12 original utterances of high-quality that served as control. As we can see in the Appendix Survey Table, the control scores stayed high and therefore, there was no reason to invalidate any participant's annotations.

14 participants (NLP and ML interns and colleagues, with no links to this work) volunteered to participate in this unpaid survey and consented verbally to the use of their data within the scope of this research. To avoid a decrease in annotation quality (due to fatigue), we split the participants in 2 groups (of 7 members) and divided the utterances in two sets (each with 60 operator-altered + 12 control utterances). We estimated the survey time to be 30-60 minutes, which was not far from the actual time (27-53 minutes).

For each utterance, we asked the participants to evaluate the intent and slot labels as *reasonable* or *unreasonable*.

| | Group 1 | | Group 2 | |
|---|---|---|---|---|
| | Experiment | Control | Experiment | Control |
| Slot | 94.5 | 94.0 | 93.8 | 97.0 |
| Intent | 89.0 | 97.6 | 85.9 | 97.5 |

Table 3: Survey results and statistics per group. All scores appear as percentages.

In Table 3 we observe a sizable decrease on the experiment side for Intent, which can be partially explained by disposition of some operators to alter a words (such as verbs) that are highly associated with the intent classification. We also observe that the Slot labeling results are high and very close to the control scores. This indicates that (contrary to many DA strategies) the NATURE operators maintain a close-to-ground-truth slot labeling.

## 5.2 Quantitative Evaluation

Table 4 show the performances of the Stack-Prop+BERT and Bi-RNN models trained on the original train data of ATIS, SNIPS and NLU-ED benchmarks. Models are evaluated on the Original, Rand and Hard test sets. We also show the scores on 10 test sets, each altered with a single NATURE operator altered test sets, where one operator is applied to the whole test set. For each benchmark, we report the F1 and accuracy on the SF and ID tasks respectively, and our End-to-End (E2E) metric. Furthermore, we report the unweighted average (Avg. column) of the aforementioned scores on the three benchmarks. Altered test sets results are sorted in descending order according to the averaged E2E metric. We notice that BERT-based models outperform RNN ones not only on original, but also on all test set variants. More precisely, we observe a gap of 6.3%, 8.7% and 5.9% on the *Avg.* E2E metric on the Orig, Rand and Hard test sets.

First, we observe a noticeable lowering in the scores on Rand, and quite a radical change on Hard test set. We must consider the possibility that the hard test set incorporates more noise than the random test sets, and this could be the cause of this low score. Depending on the benchmark, the sharpest operators are not always the ones expected to be most disruptive. Yet, the decrease in score is extreme across all benchmarks and for both models.

Second, we notice that not all operators are equally disruptive. Models seem to handle well Filler operators (except for EOS), suggesting some syntax-level pattern independence and indicating that the models are using the position of the tokens instead of the tokens themselves achieve the correct predictions. The Synonymy operators, specially the adjective and adverb, greatly deteriorate the performances. This decrease in score shines a light on the importance of the token-level pattern, signaling that the models are using certain adjectives and adverbs to make their predictions. Since adjectives and adverbs are much less diverse than the nouns and verbs, we infer that the models are using these words as prediction clues. The Speako operator is not very disruptive either, suggesting a good capacity of the models to overcome these variants and generalize using the remaining context. Interestingly, we notice that the drop of performances is highly strong on the E2E metric. For instance, using the Stack-Prop+BERT model on the ATIS test set, altered with the EOS Filler operator, we observe a 0.3% and 6.8% drop on SF and ID respectively but a 32.1% drop on E2E. We argue that E2E is a more reliable metric compared to reporting ID accuracy and SF F1 scores separately. Specially in an industrial environment, where a Virtual Assistant can only execute the

| Test Set | ATIS | | | SNIPS | | | NLU-ED | | | Avg. | | |
|---|---|---|---|---|---|---|---|---|---|---|---|---|
| | Slot (F1) | Intent (Acc) | E2E (Acc) | Slot (F1) | Intent (Acc) | E2E (Acc) | Slot (F1) | Intent (Acc) | E2E (Acc) | Slot (F1) | Intent (Acc) | E2E (Acc) |
| Stack-Prop+BERT | | | | | | | | | | | | |
| Orig | 95.7 | 96.5 | 86.2 | 95.0 | 98.3 | 87.9 | 74.0 | 85.1 | 67.8 | 88.2 | 93.3 | 80.6 |
| Rand | 91.3 | 95.0 | 66.5 | 83.4 | 96.1 | 53.8 | 67.4 | 76.1 | 56.8 | 80.7 | 89.1 | 59.0 |
| Hard | 82.3 | 90.7 | 34.9 | 70.6 | 95.3 | 12.9 | 55.5 | 62.7 | 38.9 | 69.5 | 82.9 | 28.9 |
| Pre-V. Filler | 95.6 | 96.5 | 85.6 | 92.2 | 98.3 | 79.3 | 71.0 | 83.6 | 65.7 | 86.3 | 92.8 | 76.9 |
| Syn. StopW | 93.0 | 94.8 | 76.5 | 89.7 | 96.7 | 74.3 | 70.2 | 78.9 | 60.2 | 84.3 | 90.1 | 70.3 |
| BOS Filler | 95.6 | 96.2 | 85.8 | 86.5 | 97.1 | 54.9 | 72.5 | 80.8 | 63.9 | 84.9 | 91.4 | 68.2 |
| Post-V. Filler | 94.0 | 96.5 | 80.3 | 84.8 | 98.0 | 57.1 | 68.0 | 84.1 | 63.6 | 82.3 | 92.9 | 67.0 |
| Syn. V. | 90.1 | 95.3 | 63.6 | 88.4 | 95.1 | 66.7 | 68.5 | 74.2 | 56.5 | 82.3 | 88.2 | 62.3 |
| Speako | 92.9 | 92.7 | 72.5 | 77.9 | 94.6 | 45.3 | 69.5 | 74.2 | 57.6 | 80.1 | 87.2 | 58.5 |
| Syn. Any | 90.3 | 90.5 | 54.4 | 86.9 | 94.4 | 61.6 | 67.8 | 71.0 | 53.5 | 81.7 | 85.3 | 56.5 |
| Syn. Adj. | 84.7 | 92.7 | 42.4 | 78.2 | 95.4 | 44.4 | 60.2 | 69.7 | 47.2 | 74.4 | 85.9 | 44.7 |
| Syn. Adv. | 88.2 | 89.1 | 43.9 | 77.6 | 94.3 | 41.9 | 61.6 | 65.6 | 45.4 | 75.8 | 83.0 | 43.7 |
| EOS Filler | 88.9 | 96.3 | 54.1 | 72.1 | 97.7 | 13.1 | 63.9 | 78.0 | 53.6 | 75.0 | 90.7 | 40.3 |
| Bi-RNN | | | | | | | | | | | | |
| Orig | 94.9 | 97.6 | 84.7 | 89.4 | 97.1 | 76.6 | 66.4 | 80.9 | 61.7 | 83.6 | 91.9 | 74.3 |
| Rand | 89.9 | 94.3 | 61.8 | 75.6 | 94.1 | 39.0 | 60.6 | 70.8 | 50.1 | 75.4 | 86.4 | 50.3 |
| Hard | 79.9 | 92.0 | 27.6 | 62.4 | 92.9 | 7.0 | 49.6 | 58.8 | 34.4 | 64.0 | 81.2 | 23.0 |
| Pre-V. Filler | 94.7 | 97.3 | 82.2 | 84.6 | 96.4 | 60.0 | 63.3 | 80.1 | 59.3 | 80.9 | 91.3 | 67.2 |
| Syn. StopW | 90.6 | 94.7 | 72.7 | 80.5 | 95.4 | 56.4 | 62.3 | 73.2 | 52.7 | 77.8 | 87.8 | 60.6 |
| BOS Filler | 80.7 | 96.7 | 82.6 | 80.9 | 96.7 | 38.4 | 65.8 | 78.8 | 59.6 | 75.8 | 90.7 | 60.2 |
| Post-V. Filler | 93.8 | 96.9 | 80.3 | 77.9 | 96.6 | 37.4 | 62.6 | 79.3 | 56.6 | 78.1 | 90.9 | 58.1 |
| Syn. V. | 87.6 | 95.9 | 56.6 | 79.5 | 92.1 | 50.6 | 61.3 | 70.5 | 50.7 | 76.1 | 86.2 | 52.6 |
| Speako | 91.8 | 90.3 | 68.1 | 70.1 | 90.1 | 33.6 | 61.5 | 69.8 | 51.0 | 74.5 | 83.4 | 50.9 |
| Syn. Any | 89.2 | 90.4 | 52.6 | 77.8 | 91.4 | 40.6 | 62.0 | 67.3 | 49.1 | 76.3 | 83.0 | 47.4 |
| Syn. Adj. | 81.7 | 94.2 | 34.4 | 71.7 | 93.9 | 34.9 | 54.3 | 65.5 | 42.1 | 69.2 | 84.5 | 37.1 |
| Syn. Adv. | 87.2 | 85.1 | 38.4 | 69.9 | 92.1 | 29.0 | 54.7 | 61.4 | 40.3 | 70.6 | 79.5 | 35.9 |
| EOS Filler | 88.9 | 96.8 | 52.2 | 64.1 | 94.1 | 5.9 | 56.4 | 65.8 | 42.0 | 69.8 | 85.6 | 33.4 |

Table 4: SF, ID and E2E performances of BERT and RNN based models trained on ATIS, SNIPS, and NLU-ED and evaluated on their original and altered test sets. We show results on *per-operator* as well as on Rand and Hard test sets. Furthermore, we report the unweighted average score on the 3 benchmark we considered. The lowest scores in each column appear underlined.

correct command if the intent and all slots are correctly predicted.
Additionally, to better understand the underlying processes of the state-of-the-art models, we produced and analyzed the self-attention weight heat-maps. This allows us to better understand what tokens the models focus on more to make their prediction. In Figure 4 we show a representative excerpt heat-maps for wrongly predicted sentences (for both SF and ID). One for the unchanged SNIPS test set and one for each type of operator. We observe that the self-attention often focuses more heavily on verbs, nouns and certain types of stop words, such as *"the"*. It also shows that high attention is given to verbs and certain stop words at the end of the sentence. This is evident in all Figures but particularly in Figure 4b, where we can see high attention on non-frequent tokens (for the benchmark), such as *"if"* or *"?"*.

| Test Set | ATIS | | SNIPS | | NLU-ED | | Avg. | |
|---|---|---|---|---|---|---|---|---|
| | w/o | w Aug. | w/o | w Aug. | w/o | w Aug. | w/o | w Aug. |
| Orig | 86.2 | 83.3 (-2.9) | 87.9 | 85.3 (-2.6) | 67.8 | 66.2 (-1.6) | 80.6 | 78.3 (-2.3) |
| Rand | 66.5 | 69.2 (+2.7) | 39.0 | 48.2 (+9.2) | 56.8 | 56.7 (-0.1) | 54.1 | 58.3 (+4.2) |
| Hard | 34.9 | 54.0 (+19.1) | 12.9 | 27.1 (+15.2) | 38.9 | 40.7 (+1.8) | 28.9 | 40.6 (+11.7) |

Table 5: End-to-End (E2E) scores of Stack-Prop+BERT models trained on ATIS, SNIPS and NLU-ED original (w/o) and augmented (w) training data. Each model is evaluated on its respective original, Rand, and Hard test set. We report the unweighted average of the 3 datasets.

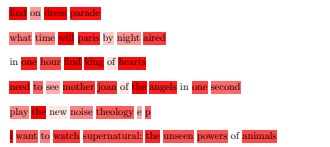

(a) Heat-map of original SNIPS utterances.

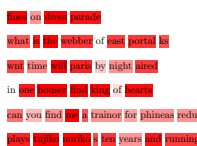

(b) Heat-map of EOS filler-altered utterances.

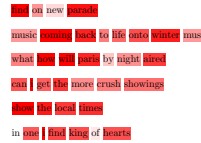

(c) Heat-map of Synonymy Adjective-altered utterances.

(d) Heat-map of Speako-altered utterances.

Figure 4: Heat-maps of SNIPS utterances whose SF and ID labels were wrongly predicted by the Stack-Prop+BERT model. The more intense the color, the greater the self-attention weight.

So far, we have shown that state-of-the-art SF and ID models do suffer when small perturbations are introduced to the test data. We now run experiments on augmented data in order to test the models' performances on larger and slightly more diverse train sets (Section 4.4). Table 5 reports E2E scores of Stack-Prop+BERT [10] model when trained without (w/o) and with (w Aug) data-augmented train and validation sets. Similar to Table 4, we evaluate the model on the Original, Rand, and Hard test sets of ATIS, SNIPS and NLU-ED while also reporting the unweighted average score.

On one hand, we observe significant gains on the altered test sets (except on NLU-ED Rand) across all benchmarks. The largest increase in performances are obtained on the Hard sets with 19.1% and 15.2% of gain on ATIS and SNIPS respectively. The gain can be partially explained by the augmentation of training data size, forcing the model to better generalize and also to the fact that our operator shares some characteristics with the used DA toolkit (i.e., Synonymy).

On the other hand, the performances decrease on the 3 benchmark, by an average of 2.3%, when the model is evaluated on the Original test sets. DA is a valid strategy in NLP, specially for small sized datasets. However, even the large and more diverse NLU-ED benchmark shows only small improvement and does not solve the unobserved pattern problem exemplified by the NATURE operators. This is a strong indicator that the problem is far from solved, and that there is much room for research.

# 6 Conclusions

Neural Network models have a black-box architecture that makes it hard to discern when they correctly generalize over the input and when they resort to heuristic features that correlate to the expected output. We present the NATURE operators, apply them to test sets of standard spoken language oriented benchmarks and observe a consequential drop of the state-of-the-art model scores. The different operators in our framework help discern what surface patterns is the model misusing. We apply simple DA techniques (that are distinct from our operators) to the train and validation sets of each benchmark, allowing us to determine when and to what extent the problem is due to a small training set size. Although DA strategies tends to improve the generalization score, they do not fully recover nor catch up to their original scores.

In future work, we expect to improve the current operators and include more diverse and realistic speech handicap, vocabulary, syntax, and miscellaneous pattern operators.

# 7 Acknowledgments

We are grateful to the participants of the survey.

---

[10]Performances of the Bi-RNN model show very similar trends.

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
