# OpenReview forum: "NATURE: Natural Auxiliary Text Utterances forRealistic Spoken Language Evaluation"
_NeurIPS.cc/2021/Track/Datasets_and_Benchmarks/Round1 — Submitted to NeurIPS 2021 Datasets and Benchmarks Track (Round 1)_

### Official Review · Reviewer_Uz16 · 2021-07-01
**New test benchmarks to evaluate the robustness of Intent Detection models against possible corruptions in text due to variations in the spoken language. Some analysis on how well this synthetically curated test set reflects the real scenario is missing. Not sure if the code, data and the test sets will be released as it was mentioned that: “The code and the data are proprietary”.**

**Rating:** 6
**Confidence:** 2
**Correctness:** The claims by the authors are correct.

**Strengths:**

The new test set is well-motivated, it is helpful to benchmark Intent detection models and evaluate their robustness for real-world applications. The proposed data augmentation technique also improves the robustness of the state-of-the-art approaches.

**Weaknesses:**

One of the key aspects that is not evaluated is how well the new test set reflects the real-world scenario. Since the dataset is automatically curated, does it cover most of the cases that can happen in the real world? I did not find any discussion in the paper as to why the proposed corruption techniques represent real-world cases well.

**Additional Feedback:**

Please check the Weaknesses section.

**Clarity:**

The paper can be improved by describing the tasks such as intent detection, slot-filling in a couple of lines. This is crucial for understanding the paper, and these terms (especially the second one) are not common knowledge.

Most of the footnotes can be part of the main text.

**Documentation:**

The details on how to create such a test set are available, but the code, data might not be released as they are proprietary

**Relation To Prior Work:**

Yes

**Summary And Contributions:**

Final rating: Initial rating was "Below acceptance threshold", have changed it now to "above acceptance threshold". Explanation in the comments of this thread.

The paper argues that the state-of-the-art Intent detection models are not evaluated on sufficiently realistic samples that occur often in real-world applications. The spoken language introduces corruptions in the text (e.g. filler words) and models need to be robust to such corruptions.

The paper shows that the state-of-the-art models are not robust and suffer a drop in performance under such corruptions. The texts in the existing datasets are automatically altered by introducing a fixed set of corruptions. The paper also introduces data augmentation techniques to improve the robustness of these models for these new test sets.

---

> ### Author Response · Authors · 2021-07-12
> **In depth-analysis currently in the making, datasets are not proprietary, code is in process of being publicly published.**
>
> Concerning Weaknesses:
> Spoken language phenomena range from accents to stutter and a comprehensive coverage in one work is challenging.  However, we base our work on some of the well-known characteristics of spoken-language (to only mention a few studies in this field, see reference below:  [1], [2], [3], [4]).  Some of these everyday characteristics are widely used in spoken-language, can be implemented in written text but rarely appear in the benchmarks.  Given the necessity of conversational agents to deal with these spoken-language particularities and their absence in the benchmarks, we thought it relevant to introduce them in a non-exhaustive manner.
>
> Concerning Clarity:
> We have added a short description of these tasks. We have made a revision of our footnotes tried to incorporate some to the main text as organically as possible. Thank you for these suggestions.
>
> Concerning Documentation:
> We have shared the datasets after modification and we are following the administrative process to37publicly publish the code on Github.
>
>
> REFERENCES:
>
> [1]    Exploring filler words and their impact (Duvall et al. 2014)
>
> [2]    Eight main differences between collections of written and spoken data (TillMann 1997)
>
> [3]    Errors and disfluencies in spoken corpora (Gilquin et al. 2013)
>
> [4]    Using uh and um in spontaneous speaking (Clark et Fox Tree 2002)

---

> > ### Comment · Reviewer_Uz16 · 2021-07-14
> > **Main queries answered; human evaluation would be incredibly useful**
> >
> > I am changing my rating to "just above acceptance threshold" because most of my queries are answered.
> >
> > I still believe conducting a human evaluation study will be very insightful for: (i) how well the curated test set matches the real-world cases, (ii) what are the possible future directions to bridge the gap with real-world cases further. I hope the authors include this in the next version of the paper.

---

### Official Review · Reviewer_ZLk3 · 2021-07-04
**Useful but Not Novel**

**Rating:** 6
**Confidence:** 4

**Strengths:**

The paper introduces a useful adversarial attack technique for slot-filling and indent detection tasks. The results are claimed to be the first to show the deficiency of the current research in this area, and highlight that there is a lot of room for future research.

**Weaknesses:**

The methods used to generate the adversarial examples are not novel and have been widely used in the NLP field, making the contributions a little incremental. For example, (Yin, et al.) attack sentence encoders with token substitutions, (Belinkov, et al.) attack neural translation systems with typoes, and (Jia, et al.) attack reading comprehension systems by adding semantic neutral sentences. With so many previous works, I am not surprised that NLP models are susceptible to the perturbations introduced in this work.

> Yin, Fan et al. “On the Robustness of Language Encoders against Grammatical Errors.” ArXiv abs/2005.05683 (2020): n. pag.

> Belinkov, Yonatan and Yonatan Bisk. “Synthetic and Natural Noise Both Break Neural Machine Translation.” ArXiv abs/1711.02173 (2018): n. pag.

> Jia, Robin and Percy Liang. “Adversarial Examples for Evaluating Reading Comprehension Systems.” ArXiv abs/1707.07328 (2017): n. pag.

**Additional Feedback:**

What does line 179 mean?
> So, we repeat this operation 10 times and calculate the mean score and variance.

**Clarity:**

The paper is basically well presented. It's worth mentioning that the line 400 - 4`10 should be deleted according to the instructions. For checklist 3. (c), where are the error bars in the paper? For 2. (a) (b), I don't find any theoretical results in the paper.

**Correctness:**

The authors have made a human-based evaluation on the generated testing sets to ensure the added perturbations are reasonable.

**Documentation:**

The authors have provided the details and the statistics for the datasets. A URL for the data is also offered.

**Ethics:**

The authors claimed that the work is not detrimental to people’s safety, privacy, security, rights, or to the environment. I also don't find any further related concerns.

**Relation To Prior Work:**

The authors have discussed similar works in other tasks for NLP, like NLI, co-reference resolution, and grammar error correction. On the other hand, more discussions about the adversarial attack techniques in the NLP field should be included, like those I mentioned in *Weaknesses*. For reference, check this [list](https://github.com/thunlp/TAADpapers).

**Summary And Contributions:**

The paper introduces a heuristic-based black-box attack technique on the slot filling and intent detection tasks. Specifically, three kinds of sentence-modifying operations are used, including adding semantic-neutral filters, replacing words with synonyms, and adding typos to words. Two heuristic strategies are used on the testing set, which include randomly apply operations on sentences and select the most misleading operation based on a surrogate JointBERT model. Both methods are applied on three benchmarks. Human-based evaluations are made to show the generated adversarial examples are label-preserving. Naturally trained models have shown significant decreases in the adversarial test sets. On the other hand, however, models augmented trained with the perturbed examples show a decrease in standard accuracy though enjoy better robust accuracy.

---

> ### Author Response · Authors · 2021-07-13
> **Response to Useful but Not Novel**
>
> Concerning Weaknesses:
> We have updated our related works section accordingly. The novelty in our work lies in the adaptation of adversarial examples techniques applied to the scarce ID and SF corpora.   A joint task that requires special attention to preserve the semantic labeling while altering the test set. Many of the aforementioned methods in NLU only apply their techniques to sentence level classification but it is more challenging to apply it to token level classification while maintaining the quality.
>
> Concerning Clarity:
> We have corrected the mistakes in the checklist according to your comments. Thank you for pointing them out.
>
> Concerning Relation To Prior Work:
> We have updated our related works section accordingly.
>
> Concerning Additional Feedback:
> Line 179 "By doing so, we obtain an operator-unbalanced test set. So, we repeat this operation 10 times and calculate the mean score and variance." means that using random selection of operators does not guarantee an equal distribution, to get a better average score we ran the experiment multiple times. We have updated the paper, making an effort to reformulate the sentence in a more clear way.

---

### Official Review · Reviewer_FNCx · 2021-07-06
**Interesting work, but not ready for publication**

**Rating:** 6
**Confidence:** 3

**Strengths:**

(1) The motivation of this paper is clear.
(2) They propose a framework to perturb a dataset, which may be useful for future research.
(3) Human evaluation is conducted to ensure the semantic equivalence between the altered test set and the original test set.
(4) Experiment results reveal that SOTA models are not robust enough to small perturbations to the input. Also, there are some interesting phenomena. For example, Pre-V fillers are less disruptive than other operators, which "may" suggest some syntax-level pattern independence.

**Weaknesses:**

(1) The resulting test set using the proposed framework may not be "natural ". If an utterance does not seem natural, I would expect the models to be more likely to fail. It's not clear whether it is meaningful to test a model's ability on unnatural text, given that it has been trained on natural text.
(2) Although the paper claims that its motivation is to explore the heuristic patterns exploited by different models. It lacks in-depth analysis of where the models fail and what specific patterns different models may have exploited when making their prediction. I do think that the data augmentation part could be replaced by some detailed analysis (e.g. error case analysis).


**Additional Feedback:**

I do think the whole story is a little plain, and I would expect the authors to add more insight into it through more in-depth analysis.

**Clarity:**

The paper is mostly clear.

Minor issues:
(1) Sentence in Line 150 seems incomplete.
(1) The Hard setting is not clear to me.


**Correctness:**

The dataset construction process is well-described. However, I do have the concern that the resulting text may not be natural. The experiment designs are appropriate and the evaluations are well-performed.

**Documentation:**

The dataset construction process is well-described. Although I do have the concern w.r.t. reproducibility since there is randomness when applying operators to the dataset.

**Ethics:**

There are no ethical concerns.

**Relation To Prior Work:**

Missing reference: Naik, Aakanksha, et al. "Stress test evaluation for natural language inference." (2018).

**Summary And Contributions:**

This paper proposes a framework that uses some heuristic rules to change the original test set without semantic altering. The proposed framework is applied to three benchmarks and the performance of SOTA models are measured on the altered test set. Experiment results show that even SOTA models are not robust to some perturbations.

---

> ### Author Response · Authors · 2021-07-13
> **Response to Interesting work, but not ready for publication**
>
> Concerning Weaknesses:
>
> (1) It is true that set alteration of any kind, from data augmentation to noise addition, distorts the data from its original archetype. However, concerning the natural/artificial qualities of the data, we believe that non-verbatim transcribed, grammatically corrected and cherry picked data, even by human specialists, does not represent the actual human-machine interaction in its most natural form (to only mention a couple of studies analyzing this difference, in the reference below: [1], [2]).  Spoken language differs greatly from written language, yet existing benchmarks that are meant to represent spoken language adhere to the written standard. This is most vital when the task(s) are expected to work from speech input, as it is the case for ID and SF. We were concerned with the capacity of our operators to alter the utterances excessively, which is why we specifically made sure to make only one change per sentence and to make it as undisruptive as possible. Concerning the natural/synthetic qualities of the operators, it is true that (unlike other works) we do not use error-full and error-less parallel data to form our operators. This is because our operators do not focus on imitating errors, they focus on imitating spoken-language, whose transcription has different properties and forms than the written-language and this kind of parallel data is incredibly rare. Instead of relying on synthetic tools and libraries, we resort to the expertise of specialists and human-populated tools and dictionaries to make the end result as natural as possible.
> (2) We evaluate the impact of individual operators in our work (table 4) and found that EOS fillers show a great drop in SF scores,  indicating that the models are using the position of the words instead of the words themselves to shortcut their predictions.   We also observe that SYN ADJ and SYN ADV also drop the SF score, signaling that the models are using certain adjectives and adverbs to deduce the slot and intent. Since the adjectives and adverbs are much less diverse than  the  nouns  and  verbs,  we  infer  that  the  models  are  using  these  words  as  prediction  clues. We have updated the section and we have added further analysis to the extent of the additional space available.
>
>
> Concerning Correctness:
>
> We cannot guarantee that the NATURE operators do not alter the natural properties of the datasets, however, to avoid it as much as possible we specifically made sure the operators make only one change per sentence and that this change is as undisruptive as possible.  We have also shared the datasets after modification and we are following the administrative process to publicly publish the code on Github.
>
>
> Concerning Minor issues:
>
> (1) The error is now corrected. (2) The hard setting is meant as a lower-bound score. Showing how much can the model be confused if we always choose the operator candidate that performs worst. Since we use a third non-related model (JointBERT) to choose the worst performing candidate, our assumption is that the candidate that performed poorly for one model will have a greater chance of performing poorly on other models. We have updated the description of the hard setting to make it clearer.
>
>
> Concerning Relation To Prior Work:
>
> We have updated the related work section accordingly.
>
>
> Concerning Documentation:
>
> In the script module (awaiting for approval to publish) we use random selection but we specify the seed whenever needed, therefore ensuring that the resulting sets are reproducible and deterministic.
>
>
> Concerning Additional Feedback:
>
> We have updated the section and we have added further analysis to the extent of the additional space available.
>
>
> REFERENCES:
>
> [1]    Eight main differences between collections of written and spoken data (TillMann 1997)
>
> [2]    Errors and disfluencies in spoken corpora (Gilquin et al. 2013)

---

> > ### Comment · Reviewer_FNCx · 2021-07-20
> > **Main concerns addressed**
> >
> > Thanks for the authors' response.  The authors addressed my main concern, and accordingly, I changed my score to 6.

---

### Official Review · Reviewer_hGKP · 2021-07-08
**Interesting work but needs better comparison to literature.**

**Rating:** 4
**Confidence:** 3
**Clarity:** The paper is well written.

**Strengths:**

ASR error-related tasks, while predominant for Speech-to-text tasks, they have not been extensively studied for NLU.

**Weaknesses:**

Augmentation strategies like synonym replacements are widely known in the literature of robustness. The proposed augmentations, in some form, have been introduced in earlier papers (see the comments in the related works). Discussion related to that, and comparison to those DA techniques are not provided in the experiments (Section 4.4 has some baseline DA techniques, but as highlighted in the Related works comments, DA techniques designed for ASR errors have not been rigorously compared to).

**Additional Feedback:**


1) Please fix the formatting of the related works section as it has artificial line breaks and is presently not in the format of paragraphs.


2) Figure 1 demonstrates the failure of the model from - A stack-propagation framework with token-level intent detection for spoken language understanding.  What is the reasoning behind choosing this model? Are there more recent models that might be more robust?


3) In Table 5, why does the performance drop when augmentations are introduced on the original test set?


**Correctness:**


The technical details are largely correct, with no immediate issue.


**Documentation:**

Could only find the released data. However, there was no documentation attached to it. Also, codes for generating the augmented data seems to be missing.

**Ethics:**

No issues.

**Relation To Prior Work:**


The discussion on related works is one of the weakest links in the present draft. Although some works related to generalization tests using test set alterations, and sensitivity towards superficial features have been discussed, there is no structured discussion on the efforts to create strong testbeds using strategies involving augmentation, human-in-the-loop adversarial testing sets (such as Dynabench or Genie.), etc.

Another major related area missing in the discussion is that of counterfactual augmentations. Papers like Learning The Difference That Makes A Difference With Counterfactually-Augmented Data, Evaluating Models' Local Decision Boundaries via Contrast Sets, etc. are popular works that discuss counterfactual tests (and training) sets to learn more robust and generalizable decision boundaries. With regards to task-oriented dialogs, the recent ICLR paper COCO: Controllable Counterfactuals for Evaluating Dialogue State Trackers demonstrate novel ways to automatically generate counterfactuals related to slot edits and counterfactual response generation. This paper does highlight that there is a significant overlap between slot values across training and testing sets and that models would exploit frequent patterns to get good results.

Closely related works to this draft, that are not discussed here include the empirical study on ASR errors in open-domain dialog - Are Neural Open-Domain Dialog Systems Robust to Speech Recognition Errors in the Dialog History? An Empirical Study, where the authors demonstrate that training with ASR errors teach models to be more robust to these superficial lexical errors. There are also many works that attempt to simulate ASR errors through various means, such as interpolations between linguistic and acoustic embeddings (Simulating ASR errors for training SLU systems), homophones (Improving the robustness of speech translation.), similar pronunciation (Augmenting translation models with simulated acoustic confusions for improved spoken language translation.), confusion n- gram pairs harvested from aligned ASR-reference text pairs (Data Augmentation for Training Dialog Models Robust to Speech Recognition Errors) etc. Similar augmentation strategies (already seen in the literature)  are present in the current draft, such as speako. While these works are not targeted for slot-filling and intent-detection, their observations with speech recognition should broadly transfer to these NLU tasks too. Slot substitution using synonyms has been discussed in Simple is Better! Lightweight Data Augmentation for Low Resource Slot Filling and Intent Classification, but again, the paper does not make the comparison.

More systematic robustness checks for evaluating models have been proposed in works like Beyond Accuracy: Behavioral Testing of NLP Models with CheckList, or more recently, Robustness Gym: Unifying the NLP Evaluation Landscape. As these works propose robustness checks in various NLP tasks, there should be a discussion if SLU tasks contain any peculiarities that the proposed draft specifically discusses.

The above references are a non-exhaustive list. As such, any revision should also include a detailed discussion (with comparisons) on other important related works.


**Summary And Contributions:**


The paper deals with robust evaluation of NLU in task-oriented dialog systems. They propose some augmentation strategies like synonym replacement, insertion of vocal words, and homophone replacement, etc. for making models more robust to slot filling and intent detection. Results show that these augmentations largely help curate a more challenging evaluation set mimicking real-life scenarios.

---

> ### Author Response · Authors · 2021-07-13
> **Response to Interesting work but needs better comparison to literature.**
>
> Concerning Weaknesses:
> The goal of the DA section is to show that these operators are distinct from "regular" DA techniques. The adaptation of related ASR techniques and their comparison with these and other operators would fall into the score of more extensive and future works.
>
> Concerning Relation To Prior Work:
> You are correct. Even though we are aware of some of the mentioned related works, due to space limitations and the fact that they do not cover intent detection and slot filling we did not mention them. We have revised our related works section. The human-in-the-loop strategies are promising yet for token-level classification such as slot filling it requires additional work. It would be challenging to find crowd workers with the right expertise for this task. In SF and ID there is a lack of any kind of robust or challenging evaluation of SOTA models.
> We have done our best to summarily discuss the content of the papers.
> We have updated the related work section accordingly.
>
> Concerning Documentation:
> We are following the administrative process to publicly publish the code on Github.
>
> Concerning Additional Feedback:
>
> We have changed the formatting accordingly.
>
> There has been a few more recent models developed for this joint task (e.g. in references below: [1], [2]). Although, at the time of the experiments the two models we chose were state-of-the-art. Just recently, the works of [1] have beaten them for some tasks by a small margin.
>
> It is to be expected that the basic data-augmentation applied on the training and validation sets alters some of the artifacts and patterns that the models use as heuristic clues. Data augmentation should also force the model to learn to generalize. For these benchmarks, it seems that the improvement from the generalization does not help the model as much as the artifacts did.
>
> REFERENCES:
>
> [1]    L Qin et al., A co-interactive transformer for joint slot filling and intent detection, 2021275
>
> [2]    Bhathiya et al., Meta Learning for Few-Shot Joint Intent Detection and Slot-Filling, 2020

---

### Decision · Program_Chairs · 2021-07-26

**Decision:**

Reject

**Comment:**

As pointed by reviewers, this work will require a series of additional work (better comparison to literature and justify novelty) to make it ready for publication.  I recommend rejecting the current version.